# Making rainfall features fun: scientific activities for teaching children aged 5-12 years

A. Gires[1], C. L. Muller[2], M.-A. le Gueut[3], and D. Schertzer[1]

[1]HMCo, École des Ponts, UPE, Champs-sur-Marne, France
[2]Royal Meteorological Society, Oxford Road, Reading, RG1 7LL, UK
[3]Editions le Pommier, Paris, France

*Correspondence to:* A. Gires (auguste.gires@enpc.fr)

**Abstract.** Research projects now rely on an array of different channels to increase impact, including high-level scientific output, tools and equipment, but also communication, outreach and educational activities. This paper focuses on education for children aged 5-12 years and presents activities that aim to help them (and their teachers) grasp some of the complex underlying issues in environmental science. More generally, it helps children to become familiarized with science and scientists, with the aim to enhance scientific culture and promote careers in this field. The activities developed are focused on rainfall: (a) designing and using a disdrometer to observe the variety of drop sizes; (b) careful recording of successive dry and rainy days and reproducing patterns using a simple model based on fractal random multiplicative cascades; and (c) collaboratively writing a children's book about rainfall. These activities are discussed in the context of current state-of-the-art pedagogical practices and goals set by project funders, especially in a European Union framework.

## 1 Introduction

Research projects now rely on an array of different channels to increase impact. This obviously includes high level scientific output, tools and instrumentation, but also communication, outreach and educational activities. This paper focuses on education for young children (5–12 years old) and presents a number of activities and a science book with the aim of assisting them (and their teachers and parents) to grasp some of the complex underlying issues in the field of environmental science, with a focus on rainfall. More generally it helps children to become familiarized with science and the role of scientists, with the aim of enhancing scientific culture and promoting careers in this field. The classroom activities presented form part of the dissemination effort of the NEW Interreg IV RainGain project (www.raingain.eu) and the Ecole des Ponts ParisTech Chair "Hydrology for resilient cities" endowed by Véolia (www.veolia.com). The three activities are specifically dedicated to transmitting knowledge on rainfall features:

- design and implementation of a drop measurement device;

- recording and modelling of the succession of dry and rainy days;

- collaborative writing of a scientific book about rainfall.

The activities will be discussed in the context of current pedagogical practice and goals set by project funders, especially in a European Union framework.

**A pedagogic perspective**

Good science education is essential in early childhood, not only for social and cognitive development, but also for engaging young (3-5 year old) learners with science (Sackes et al., 2009). However, science is often perceived as unappealing to young learners (Koren and Bar, 2009; Sjøberg and Schreiner, 2005; Stefansson, 2006; Muller et al., 2013). Many studies have highlighted the need to engage and enthuse learners at a young age: Tai et al. (2006) found that American students reporting an interest in science careers at the age of 13/14 were more likely to obtain a university degree in a science field than those with no interest; the Royal Society (2004) found that 63 % of their UK study participants had first considered a career in a science and engineering field by the age of 14; Maltese and Tai (2010) found 30 % of participants having an interest in middle school or high school.

A positive attitude towards science in school will often lead to a positive commitment and lifelong interest in the subject (Simpson et al., 1994). Bennett and Hogarth (2009) showed that positive attitudes to school science declined significantly between the ages of 11 and 14, whilst Lyons (2006) found that students are often not engaged by the "autocratic" way science is represented in their classes, finding that it is often disconnected from the natural world they experience on a day-to-day basis. Maltese and Tai (2010) found that early interest in science was provoked by specific memorable activities (specifically school-based experiences related to demonstrations) or an exceptional teacher.

Previous work has also found that the key to students' understanding of science is activities which actively involve the student. As such, numerous studies (e.g. Muller et al., 2013; Alrutz, 2004; Buncick et al., 2001; Cleaves, 2005; Dorion. 2009, 2007; Ellington et al., 1981; Harvard-Project-Zero, 2001; Lyons, 2006; Maltese and Tai, 2010; Odegaard, 2003; Royal Society, 2004; Osborne et al., 2003; SATIS, 1986; Sloman and Thompson, 2010; Tai et al., 2006; Tobias and Hake, 1988; Wagner, 1998) have found that including a variety of activities and methods to engage students with different interests and experiences, providing an engaging classroom environment and allowing students to feel comfortable asking questions are important factors that can invoke interest in science and foster retention.

Using hands-on activities is clearly a popular and successful method to engage students. However, many teachers find science challeging to teach and as such turn to picture books and children's science literature for assistance. Sackes et al. (2009) explored the benefits and limitations of using children's literature to introduce science concepts. The authors found that although some books are poorly written and can spread misconception, those written under the guidance of science consultants were great learning tools, offering unique opportunities for introducing science concepts in the early years, fostering interest, curiosity and positive attitudes, as well as integrating literacy and science (Broemmel and Resarden, 2006; Castle and Needham, 2007; Coskie, 2006; Monhardt and Monhardt, 2006). Pringle and Lamme (2005) found that picture books in particular were very useful for communicating concepts in a welcome and familiar format, and demonstrating logical connections that exist between classroom learning and the natural worlds outside the classroom. Thus children's books – when produced *and*

used accurately and effectively (Ford, 2006) – are a key part of supporting children's development of scientific concepts (Zeece, 1999).

The following chapters outline a number of hands-on activities and a scientific book that have been developed to support the teaching and learning of complex topics at a young age. The presentation of each activity is preceded by an historical and scientific background section and followed by a "going further" section. The purpose of these sections is to provide teachers and educators with sufficient material so that they feel comfortable when implementing the activity. According to the age of the children, they may or may not use this additional information. The activities were initially implemented in a classroom with children aged 5-6 years. This was a practical choice driven by the fact that the son of the first author was in that class, which facilitated the first contact with the teacher! Since then other implementations have been carried out. The target audience of the collection to which the book belongs is 8-12 years. Hence the collaborative shaping of the book was achieved with a classroom of children aged 8-9, i.e. the youngest of the age range targeted. This was to make sure to make sure that it is understandable for the whole age range targeted.

## 2  Drops are not all the same: the flour or oil disdrometer experiment

This activity consists of designing and testing disdrometers made from paper plates containing a few mm of flour or oil to observe rain drops individually.

The aim of the activity is two-fold:

1. learning how to design and test two instruments as well as compare their pros and cons in "laboratory" and "actual" conditions;

2. unveiling the unknown diversity of rain drop sizes and providing some basic explanations.

The activity was implemented in October 2013 in a classroom with children aged 5–6, in Sceaux (South of Paris, France).

### 2.1  Historical and scientific background

The idea of this activity is to reproduce in a more "artisanal" way a famous experiment by Marshall and Palmer (1948) who used dyed filter paper to get an estimate of the Drop Size Distribution (DSD). In the same paper, they used these observations to calibrate the famous relation $Z = a R^b$ (with $a = 200$ and $b = 1.6$) between the reflectivity $Z$ measured by weather radar (basically the power of the wave backscattered by drops in the atmosphere) and the rain rate $R$, the variable hydro-meteorologists are interested in. This relationship is still commonly used, and its establishment was a milestone in weather radar applications. A similar experiment was carried out by Lovejoy and Schertzer (1990) who used $128\,\text{cm} \times 128\,\text{cm}$ of chemically blotted paper. They showed that drop centres are not homogeneously distributed but rather exhibit some clustering with an underlying scale-invariant fractal distribution. Interestingly, rain drop fossils have also recently been used to characterize ancient DSDs and thus provide information about air density 2.7 billions years ago, when the imprints were formed (Som et al., 2012).

## 2.2 Presentation of the experiment and discussion

### 2.2.1 Design and lab test of the devices

The disdrometer is made of few mm of flour or oil in a plate (or any similar sort of medium). To understand the functioning of the oil disdrometer, children first need to notice that oil and water are not miscible. This is demonstrated using glasses containing either milk or oil, to which a drop of water is added using a pipette (Fig. 1a). Following a suggestion of the teacher the water within the pipette is colored to increase the visibility of the output. The behaviour within the two glasses is very different (Fig. 1b); in the milk, everything gets mixed, whereas in the oil, the drops remain independent of the oil and sink. In order to help children interpret and analyse the experiment, they are asked to illustrate their observations (Fig. 1c). This use of personal drawings is one of the basic ideas underlying the pedagogy promoted by the "La main à la Patte" foundation (http://www.fondation-lamap.org/en/international), and was suggested by the teacher.

The disdrometers are constructed by placing a few mm of flour or oil onto a plate. Artificial drops of colored water are dropped onto the distrometer using a pipette. Half of the children test the flour device, while the other half test the oil distrometer (Fig. 2). The children were all able to create their own device without wasting or spilling oil or flour, and only three of them needed significant help in the process. The use of a deep container as suggested by the teacher also helped! The basic premise of the session is for it to be interactive, allowing the children, the scientist and the teacher to discuss, understand and compare the functionality of the devices. Children started by comparing their device with their neighbour's one since he/she had made the other type of device. The teacher was helpful in starting the discussion, because the scientist was not used to this at the beginning.

The main learning concepts are:

– Once a droplet falls onto the flour disdrometer, it creates a small wet crater that remains visible;

– Once a droplet falls on the oil disdrometer, it does not mix with oil and remains visible where it landed;

– The flour device can be easily transported while the oil one cannot. Indeed, as soon as the device is not completely horizontal, droplets begin to move and merge when they reach the lowest part of the plate/container. This does not occur with the flour device which tolerates being slightly tilted;

– The oil disdrometer retains a better imprint of drop size. Indeed with the flour device, the water slightly spreads around the small crater. Hence the actual size is lost, and only the relative sizes are accurate. With the oil device, the shape of droplets are lost as they become spherical, however the volume remains accurate since the water of the drop and oil do not mix.

Being very visual, the first three items were easily grasped by children, whereas the fourth one was a bit trickier and required more detailed explanations and illustrations for few of them. The size differences between imprints in flour and in oil while drops were created using the same pipette was initially noted only by a few children.

### 2.2.2   Outside implementation and drop analysis

The second part of the activity consists of testing the disdrometers under actual rainfall. For this a volunteer needs to go outside with the disdrometer, uncover it for a few seconds, and return inside for analysis (Fig. 3a). Typical results are displayed on Fig. 3b. We were lucky that it was actually raining the day of the experiment. In case it had not been, some pictures were ready in order to continue the discussion anyway and the teacher would have done the actual test once some rain appeared.

At this stage it should be noted that the oil disdrometer is unsatisfactory under real conditions because when a droplet impacts -or more precisely "crashes" into- the oil surface, it brakes up into several droplets, thus biasing the results. However, the fact that a device which seemed effective during inside lab testing failed under "real" conditions is an interesting lesson for children. In order to help children notice the various sizes of drops and their inhomogeneous distribution, they are also asked to draw their observations (Fig. 3c). The variety of drop sizes was visible on about 2/3 of the children's drawings. Let us mention here that the children were not asked to measure the drops' sizes because they were too young to achieve this, which is why we used the drawing. With older children it is possible to include measurement in the activity, as done recently by Mazon and Viñas (2013) who implemented a similar low-cost "flour" disdrometer experiment with high-school students. Before going on it should be mentioned that the actual skill involved in doing a measurement is a learning goal as well. Indeed it is an empowering notion that you can know something by measuring it yourself, instead of trusting the knowledge passed on to you by others.

Once they have observed the variety of drop sizes, the children are given some insights into the formation and development of rainfall. The main elements for such a young audience are:

i. Water vapour evaporates from the Earth's surface and moves up through the atmosphere until it reaches a colder height, where it starts to condense around a small particle (known as a "condensation nuclei" e.g. dust, soot, pollutants);

ii. Droplets grow by further condensation or merging with other droplets after a random collision. An area with numerous droplets forms a cloud;

iii. Once a droplet becomes too heavy to be held in the atmosphere, it begins to fall;

iv. As the droplets fall, there are further collisions and break-ups leading to a range of droplet sizes (equivolumic diameter) typically between 0.2 and 5–6 mm at ground level, the more numerous ones being of size 1–2 mm.

For this part of the activity, no dedicated tools were used and it was only based on a discussion. To illustrate the first point, the standard example of the condensation around a bottle taken out of a refrigerator was used and some children recognized this effect. Some specific activities should be developed to address these issues in future works.

### 2.3   Going further

Similar images can be obtained using a 2D-video-disdrometer which estimates the features (size, fall velocity, and position) of the drops falling within a sampling area of approximately 11 cm × 11 cm (see Kruger et al. (2006) for a detailed description of

the device). This device enables observations - such as those obtained with the flour disdrometer - to be recorded automatically. Figure 4 displays the droplets recorded over 1s (for each plot) during an event that occurred on 24 September 2012 in Ardèche, France. Such figures can be used to further illustrate the diversity of drop sizes and the variability observed over time, and can be compared to the children's drawings. An example of the use of such data can be found in Gires et al. (2015), who computed the time needed for a given number of drops to fall through the sampling area, and showed that the distribution exhibited a power-law fall-off confirming the inhomogeneous nature of drop distribution.

## 3   Rain or no rain: a fractal perspective

This activity consists of recording a daily time series of rainfall occurrences over two months, in order for children to understand the complexity of succession of dry and wet days, and of implementing a stochastic cascade model to reproduce patterns similar to the observed ones. The activity was tested in the same classroom as for the disdrometer experiment, with children aged 5–6 years in January 2014 in Sceaux (South of Paris, France).

The aim of the activity is two-fold:

1. Assisting children to understand the difficulty of carefully recording data over a long period of time;

2. Introducing the notion of a "model", as well as "randomness", with which they are not familiar. The idea is for the learners to become involved in the concepts rather than to formalize the complexities of them, which would be difficult for them to grasp.

### 3.1   Historical and scientific background

Rainfall occurrence patterns are tricky to characterize, model and simulate at all scales and it still remains an open issue. See, for example, Gires et al. (2013) or Schleiss et al. (2014) for recent papers on cascade-based or geostatistics-based approaches. However, it is a relevant concept, given the importance of the rain/no rain intermittence. An illustration of this is the number of zeros recorded in rainfall time series. For instance Hoang et al. (2012) reported typically about 96–98 % zeros for a long (many years), high resolution (5 min) rain gauge time series over France. For practical reasons, and due to the necessary implementation of the experiment in classrooms, the activity was conducted at daily resolution, similar to Hubert and Carbonnel (1989) who analysed a 45 year daily rainfall time series of Dédougou (Burkina Faso).

A possible solution to model observed rainfall occurrences patterns is to rely on a scale-invariant multiplicative cascade framework (Lovejoy and Mandelbrot, 1895; Lovejoy and Schertzer, 1985; Hubert, 1988). Cascade models were initially developed to tackle atmospheric wind turbulence and explain how energy is transferred from scale to scale down to the dissipation scale. It was later used for rainfall, assuming that the unknown equations governing rainfall inherit the scale-invariant properties of the Navier–Stokes equations (Hubert, 2001). They remain the same after scale contraction; suggesting that similar structure will be visible at all scales. The cascade concept, formalized by Kolomogorov in 1941 and refined in 1962 (Kolomogorov,1941, 1962) was first hinted at by the so-called father of weather prediction Lewis Richardson (1922) in a foot note:

Big whorls have little whorls that feed on their velocity, and little whorls have smaller whorls and so on to viscosity – in the molecular sense.

To illustrate these cascade models, let us introduce the pedagogical discrete case, where scales are discretized (see Fig. 5a for an illustration in 1D). At the beginning we have a structure with a given uniform level of intensity (typically a rain rate). The goal is to distribute this intensity over the domain (in time here). At each step of the cascade process, a structure is divided into two sub-structures and the intensity given to a sub-structure is the one of the parent structure multiplied by a random multiplicative increment. Repeating this process yields the desired variable field. A mathematical presentation can be found in Annex A.

## 3.2 Description of the experiment and discussion

### 3.2.1 Careful recording of rainy and dry days over a period of two months

The first step of the experiment consists of recording rainy days over a long period of time and plotting the data. Over a two month period the recording of rainy and dry days was undertaken at the start of the day, during the teacher's introduction to the day's schedule. If rainfall was noticed between 09:00 LT (Local Time) on the previous day and 09:00 LT that morning, then it is considered as a rainy day. To determine whether it had rained during the night the children checked whether the ground was wet while coming to school. At the time of the experiment, the children did not attend school on Wednesdays, Saturdays and Sundays, thus they alternatively volunteered to be responsible for recording this information on each of these days. If the teacher resides near to the school, they can also record the observations during holiday period (during which children obviously do not attend school), otherwise it is simply considered as "missing data". A bar time series was used to graphically represent the data, with each bar representing a day. The time series obtained that year is displayed Fig. 6. Black bars correspond to rainy days and white ones to dry days.

It is important to use this time to raise the children's awareness of the time and effort needed to collect and properly record data over a long period. This is often difficult and not really gratifying (or at least recognized) work. Yet it is essential to scientific research and the quality and robustness of the obtained results rely on its proper realization. This is a practical way for learners to understand and comprehend what "scientific research" consists of, and the role of the research scientist. They are essentially acting as researchers for the duration of this activity.

### 3.2.2 Modelling the succession of dry and rainy days

The second stage of this activity consists of implementing the cascade model that will enable the children to reproduce patterns similar to those they observed on their own recorded time series. It is a way to smoothly introduce the notion of a model. The word "model" itself is actually not mentioned in the class since it is too abstract for them to understand. The idea is simply to have them notice that while implementing a "recipe", they are able to generate time series that look like their observations. It also enables us to introduce the notion of randomness.

The suggested model is displayed Fig. 5b. It is actually an activity designed to imitate a $\beta$-model (see Appendix A) that can be implemented by 5–6 year old children, which was problematic to achieve. Each child is given the scheme with empty boxes (all white) and asked to follow these steps:

i. Filling the boxes: each child is given a dice with either four or five black sides; the remaining side(s) are white. For each box, the child throws the dice; if a black side is obtained, they color the box black, otherwise it remains white (see Fig. 7). This mimics the generation of the random multiplicative increments $\mu\varepsilon$ mentioned in Sect. 3.1 and discussed in Annex A. Black corresponds to "alive" (or rainy) and white to "dead" (or dry) with the naming explained in Annex A.

ii. Generation of the time series: for each box of the time series (bottom part of Fig. 5b) which correspond to a day, the child follows the line up to the upper box at the top of the scheme. If they encounter a white box, then they leave it white as a dry day. If all the boxes are black, then the day is denoted rainy and it is colored black. This process actually mimics the multiplicative process Eq. (A2) taking advantage of the fact that a multiplication by zero yields zero as output anyway. It means that a $\beta$-model is actually implemented to generate a 16 day time series. Once completed, the children cut out the time series and line them up to obtain a longer one.

Figure 8 displays examples of time series generated by children with either four or five black sides on their dice. As expected this activity was trickier to implement than the disdrometer one which is more "hands-on". On the one hand, the first part, i.e. throwing dice and coloring boxes in black or white, went smoothly. On the other hand the generation of the time series turned out to be more complicated. Most children did not understand the process of how the time series should be filled from the black and white boxes with the group explanation. Hence a one-to-one explanation using supporting examples (doing with them the firsts time steps) was needed before they were able to do it. For future implementations a group explanation might be tried but with an example done thoroughly. It also appeared that the activity was too long for the limited concentration capacity children have at this age. Hence it would be a good idea to use a three-level model rather a four-level model so that the activity is shorter.

Once completed, a discussion about whether the simulated time series exhibited patterns similar or not to the observed series took place (Fig. 6). It is important for them to understand that although the series they obtained are not the same – since the outcome of throwing a dice and therefore their boxes colors are random – the patterns are similar because the same underlying process was used. Although it is difficult to know how much they grasped of this, they all noted that time series produced with a four-black-sided dice were much drier than for the other dice. It was concluded that simulations looked like observations for the dice with five black sides and much less for the four-black-sided dice. This is in agreement with expectations (see Appendix for details). It is likely that the interpretation of this activity was too complicated for such young children and it should be tested with older ones.

## 3.3 Going further

Only rainfall occurrence was addressed in this activity, meaning the complex rainfall process was reduced to the oversimplifying binary question of rain or no rain. It is not the case in reality, since the intensities observed during rainy periods are extremely variable over a wide range of spatio-temporal scales.

Actually the $c$ parameter of the $\beta$-model (or the proportion of black sides in the dice) can be interpreted as the fractal co-dimension of the geometrical set made of the portion of time where some rain was recorded. This notion characterizes in a scale-invariant way the space occupied by a geometrical set. It appears that this fractal co-dimension depends on the threshold used for defining the occurrence or not of rainfall. Indeed when increasing the threshold, the support gets smaller and the fractal co-dimension increases (Lovejoy et al., 1987; Hubert et al., 1995). It means that in order to characterize and model an actual rainfall time series, an infinity (one per threshold) of fractal co-dimensions is needed. This is an intuitive (not mathematically rigorous) way of understanding multifractal fields, which is a framework enabling the analysis, modelling and simulation of fields that are extremely variable over a wide range of scales, such as rainfall.

These notions were not addressed with 5-6 year old children, who were too young to grasp them. But if a time series with amount is recorded with older children (using simple rain gauges), it is possible to introduce this. One should plot the rainfall occurrence pattern in a bar time series as done here, and repeat the exercise only with the day when rainfall exceeded a given threshold. If the threshold is carefully chosen, the series generated with the dice with five black sides will be similar to the initial observations, whereas the ones obtained with a dice with four black sides will be similar to the thresholded one.

## 4 Writing a scientific book on rainfall with and for children aged 8–12 years

This activity involves writing a scientific book for children aged 8–12 years, based on questions they raised themselves. It was tested in a class with children aged 8–9 years in Sceaux (South of Paris, France) in October and November 2014. The book was published in February 2015 (Gires, 2015).

The process leading to this book was designed by the editor of the "Minipomme" (Ed. Le Pommier) collection in which it was published. It is split into three main successive steps:

  i. A 1.5 h interactive session with the scientist and a class of 8–9 year old children. They were given the general topic (in this case, rainfall) of the book a few hours prior to the session and asked for any questions they had about the topic. The topic of a book in this collection should be related to some aspects of the children's curriculum, meaning that they are not completely unaware of the topic. In the case of this book about rainfall, they had recently studied the water cycle, and were already aware that water can exist in its three states (solid, liquid and gas) on Earth. This facilitated the discussion. The session was designed as an interactive session, meaning that it was more than a simple questions and answers session. Indeed the scientist did not directly give the answers, but tried to encourage the children to think about the process and suggest some answers themselves before providing a more precise explanation. To illustrate this point, they were for example asked what happens when a lot of droplets are in a cloud and move randomly, and concluded

themselves that if the droplets are too numerous they start to collide and merge, which slowly gives rise to drops that will ultimately fall. In the explanation, the scientist also used a lot of images to connect the new knowledge to existing knowledge or common experiences. What happens when you empty a bottle of water on impervious ground when it is hot and sunny? The water evaporates. The example of the cold bottle around which water condenses was again used, and here all the children had already noticed that, which was not the case for younger ones. To illustrate how small droplets are maintained in the atmosphere because of the small scale turbulent wind, the dust visible in the air when a sun ray is entering through a window was used. All children had already seen that effect.

Naturally some of the questions raised were surprising and unexpected, in which case the scientist went on to research the question in more detail before providing an answer during the second session. The two most striking examples in this specific case were "What is the taste of rain?" and "Should I walk or run under the rain to get less wet?". The latter was especially fun, and after a little research it turns out that almost ten papers based on numerical or actual experiments can be found on this topic in the scientific literature (see Bocci (2012) for a recent study with many references within). It appears that in general one should run as fast as possible when it is raining, but in some windy conditions or for certain body shapes, there exists an optimal velocity.

ii. The book is then written by the scientist, with the aim of answering (at least partially) all the questions raised by children. It is made of two parts:

- First, a lively story, involving discussions between a few characters, which contains most of the scientific elements. The story should be more than a simple dialogue; a genuine fiction should take place so that children do not even notice they are learning and grasping new concepts. As pointed out by Dahlstrom (2014) and Hut (2016), using narrative and storytelling can enhance the efficiency of geoscience communication. The story developed was based on the random and fortunate meeting of two young children with a "rain explorer" who takes them onboard her "drop's vessel" for a journey into the clouds. The story is structured around four main questions: (a) How do you measure rainfall? (b) Does it rain the same everywhere and all the time? (c) How are droplets formed? and (d) What happens when drops fall? In addition there are a few sidebars for additional details on difficult topics or definitions of complex words.

- Second, a section that includes some components for children as well as their parents (here three topics are addressed: rainfall radar measurement, lighting and thunderstorms, three states for water), and some suggestions of experiments so that children can put in practice the newly acquired knowledge either at home or in classrooms. The experiments consist of the design and testing of a flour disdrometer (see Sect. 2), and the building of a simple rain gauge out of a plastic bottle.

iii. A 1.5 h interactive feedback session: the draft of the book was given to children for reading a few days before a feedback session with the scientist was held. The main point was to ask them if they had understood everything, and whether they had some suggestions regarding to the characters. They were happy with the characters and had only minor suggestions

for the content. For example the explanation of a rainbow effect was re-written. Actually they had more questions on the process of book creation (how many people worked on it ? How long does it take ? How is it printed ? ...), which the editor answered. After this session, the scientist made some minor adjustments to improve the book.

iv. Illustration: finally, the book was illustrated by a professional designer, with a scientist providing precise schemes for drawings involving scientific content.

The book is then made available to the public (bookshop, internet...). Typically 2000 to 3000 are sold in this collection over the life of the book. It has not yet been translated to other languages. The authors did not receive very precise feedback from the teacher apart from the fact that they were satisfied with the experience. The two interactive sessions were dynamic, which shows the interest of the children in the activity.

## 5   Conclusions

In this paper we have presented various hands-on activities for young children, designed to help them become familiarized with some complex notions associated with rainfall in a playful way. They designed a device whose main purpose was to record drop size; implemented it and observed the variety of drop sizes. They also carefully recorded the succession of dry and rainy days over a period of two months before reproducing observed patterns with the help of a random fractal cascade model. Finally they helped to shape the content of a scientific book about rainfall. The goal of these activities was not only for them to acquire knowledge on the specific topic of rainfall but also to become familiarized with science and the scientific approach; to become curious about their surroundings, to develop a willingness to observe more precisely their environment, to notice details, and ultimately to begin asking questions.

The development of these activities highlighted the importance of a genuine collaboration between scientists and school teachers, which turns out to be necessary for a successful implementation. The scientist brings the initial ideas and the expert knowledge for accurate science, and makes sure that simple explanations are not simplistic and biased ones. The school teacher helps in adapting the language for young children, and also in shaping the activity so that it fits into the classroom habits so the children are comfortable with it.

Finally it should be mentioned that the activities done with children aged 5-6 years were reported on the class' blog maintained by the teacher. This is a non-public blog accessible only to the parents because it contains pictures of their children while they are at school. Since the activities were implemented in the class of the first author's son, the publication triggered some unusual and pleasant discussions on drop size distribution at the end of the schoolday. Indeed parents were not aware of the variety of drop sizes and wanted to learn more on this issue. Some of them actually tested the experiment at home with their child. The first author used these short discussions to make parents aware of the extreme variability of rainfall as well as the difficulty of properly measuring it. This reflects nicely how an activity designed for young children actually ends up with parents learning new things as well.

Future work will involve the development of more activities on rainfall to ultimately create a whole activity kit on this topic. It will also be necessary to set up an appropriate protocol that enables a quantitative evaluation of the activities, both in terms

of knowledge on the specific topic of rainfall and children's engagement with science. This will require collaboration with pedagogical experts as well.

## Appendix A:  The $\beta$ cascade model

The purpose of this appendix is to provide the interested reader with some mathematical details about cascade processes and describe the simple $\beta$ model. The process is illustrated in Fig. 5a. At the beginning the activity ($\varepsilon_0 = 1$) it is uniform over a structure (a $d$-dimensional cube, $d = 1$ for the time series studied here) of characteristic length $L$ ($\lambda = 1$). One step of the cascade process consists in breaking each structure into smaller ones with a scale ratio $\lambda_1$ (larger than one and usually equal to 2 although it is not mandatory). As a consequence after $n$ steps, there are $\lambda_1^{dn}$ sub-structures of characteristic length $l_n = \frac{L}{\lambda_1^n}$. The resolution of the process, which is the ratio between the outer scale ($L$) and the observation scale ($l_n$), is then equal to $\lambda = \frac{L}{l_n} = \lambda_1^n$. The activity $\varepsilon_n$ (i.e. $\varepsilon_{n,i}$, with $i = 1, \ldots, \lambda^d$) given to a daughter structure is equal to its parent's one multiplied by a random variable ($\mu\varepsilon$): $\varepsilon_n = \mu\varepsilon\,\varepsilon_{n-1}$. Building a cascade process basically requires determining: (i) how to divide each structure into sub-structures, (ii) the probability distribution of the random multiplicative increment. The key assumption is that these two properties are the same at all scales. The probability distribution of the random increments should be chosen so that $\langle\mu\varepsilon\rangle = 1$ to ensure ensemble conservation through scales.

Numerous models have been suggested in the literature and only the simplest one will be discussed here since it will be implemented within a classroom environment. It is often called the $\beta$-model (Frisch et al., 1978; Mandelbrot, 1974; Novikov and Stewart, 1964) and assumes that structures are either dead (inactive) or alive (active). In this model, the multiplicative random increments $\mu\varepsilon$ only have two possible states, whose probabilities of occurrence are defined by:

$$Pr\left(\mu\varepsilon = \lambda_1^c\right) = \lambda_1^{-c} \text{ (alive)} \tag{A1a}$$

$$Pr(\mu\varepsilon = 0) = 1 - \lambda_1^{-c} \text{ (dead)} \tag{A1b}$$

where $c$ is a parameter of the model. The words "dead" and "alive" are the ones historically used in the literature for this model, and they are kept here although more appropriate ones would be "dry" and "rainy" in the context of this paper. The value affected to the boost $\mu\varepsilon = \lambda_1^c$ ensures conservation of the average activity $\varepsilon$ (i.e. $<\mu\varepsilon> = 1 \Leftrightarrow <\varepsilon_n> = <\varepsilon_0>$ where $<>$ denotes ensemble average). At each step of the process the fraction of alive structures decreases by a factor $\beta = \lambda_1^{-c}$, and their activity is increased by the factor $1/\beta$ to ensure (average) conservation. After n steps of the process, i.e. at a resolution $\lambda_n = \lambda_1^n$, the sub-structure activity (equal to the product of the successive random increments)

$$\varepsilon_n = \varepsilon_0 \prod_{i=1}^{n} (\mu\varepsilon)_i \tag{A2}$$

exhibits two possible states, dead or alive, with the probability of occurrence:

$$Pr\left(\varepsilon_n = \lambda_n^c\right) = \lambda_n^{-c} \text{ (alive)} \tag{A3a}$$

$$Pr\left(\varepsilon_n = 0\right) = 1 - \lambda_n^{-c} \text{ (dead)}. \tag{A3b}$$

Such model was for instance employed by Over and Gupta (1996) or Schmitt et al. (1998) in a continuous version to represent rainfall occurrence patterns.

For practical implementation in the children's activity, we typically have $c \approx 0.3$ over the Paris area on scales ranging from one to 16 days, which yields to a probability of an "alive" random increment equal to $\lambda_1^{-c} \approx 0.81$. This value is actually very similar to $5/6 \approx 0.83$ found with the five-black-sided dice used in the activity.

**A1**

*Acknowledgements.* A. Gires would like to thank his two sons Nathanaël and Nikita for always volunteering to test new scientific experiments and helping to improve them ! The authors would like to thank V. Rouelle and D. Bourdin for opening their class at the Ecole des Clos Saint-Marcel in Sceaux (France) to scientific experiments and fruitful discussions that helped improve the activities. The authors from Ecole des Ponts ParisTech greatly acknowledge partial financial support form the Chair "Hydrology for Resilient Cities" (endowed by Veolia) of Ecole des Ponts ParisTech and EU NEW-INTERREG IV RainGain Project (www.raingain.eu). The authors would like to thank Tim Raupach for his help in proofreading the final version of the paper.

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

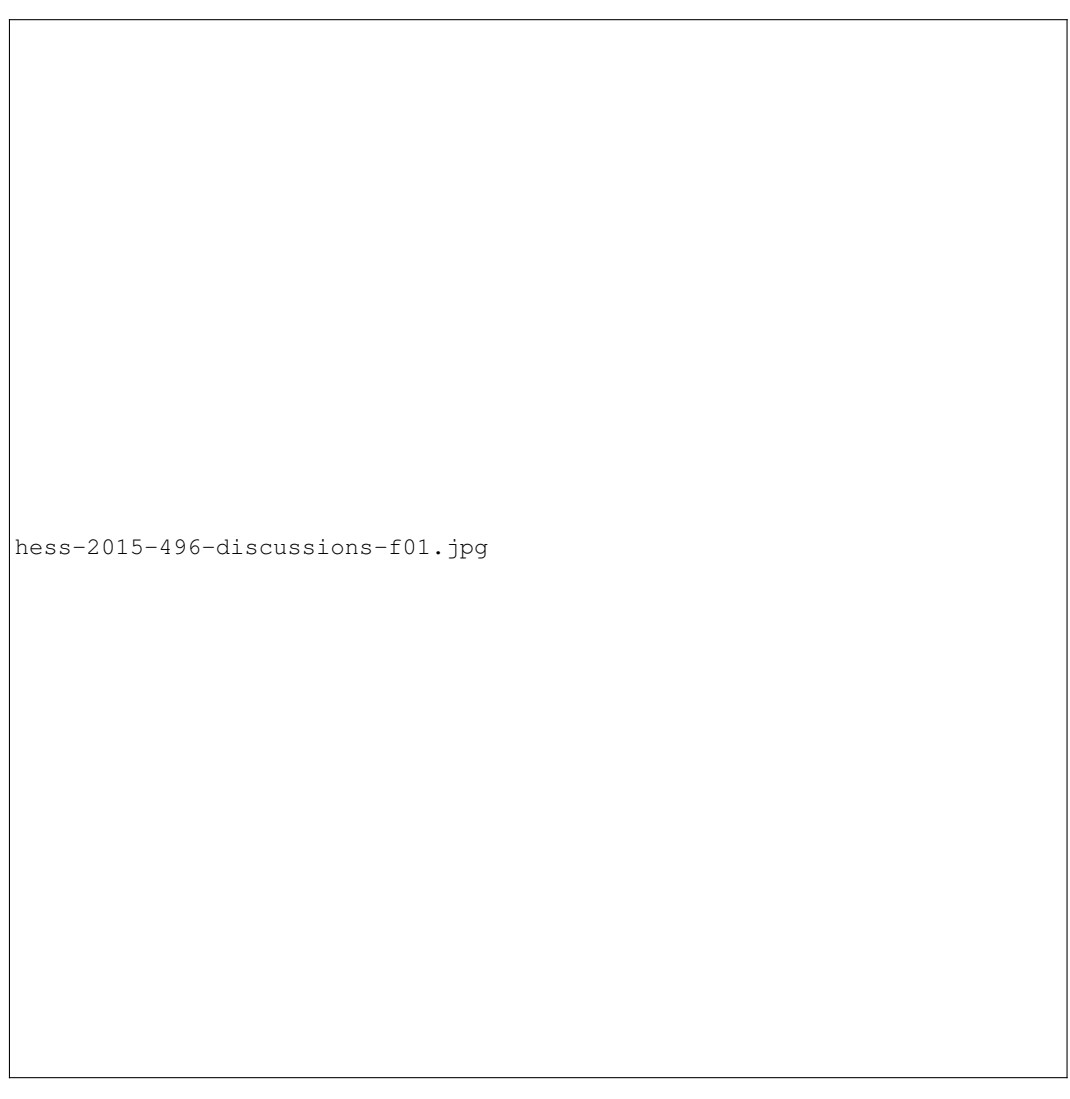

**Figure 1.** Illustration of the fact that water and oil are not miscible. **(a)** Adding colored water drops with a pipette in a glass of oil. **(b)** Outcome of the experiment with milk (left) or oil (right). (c) Child drawing observations.

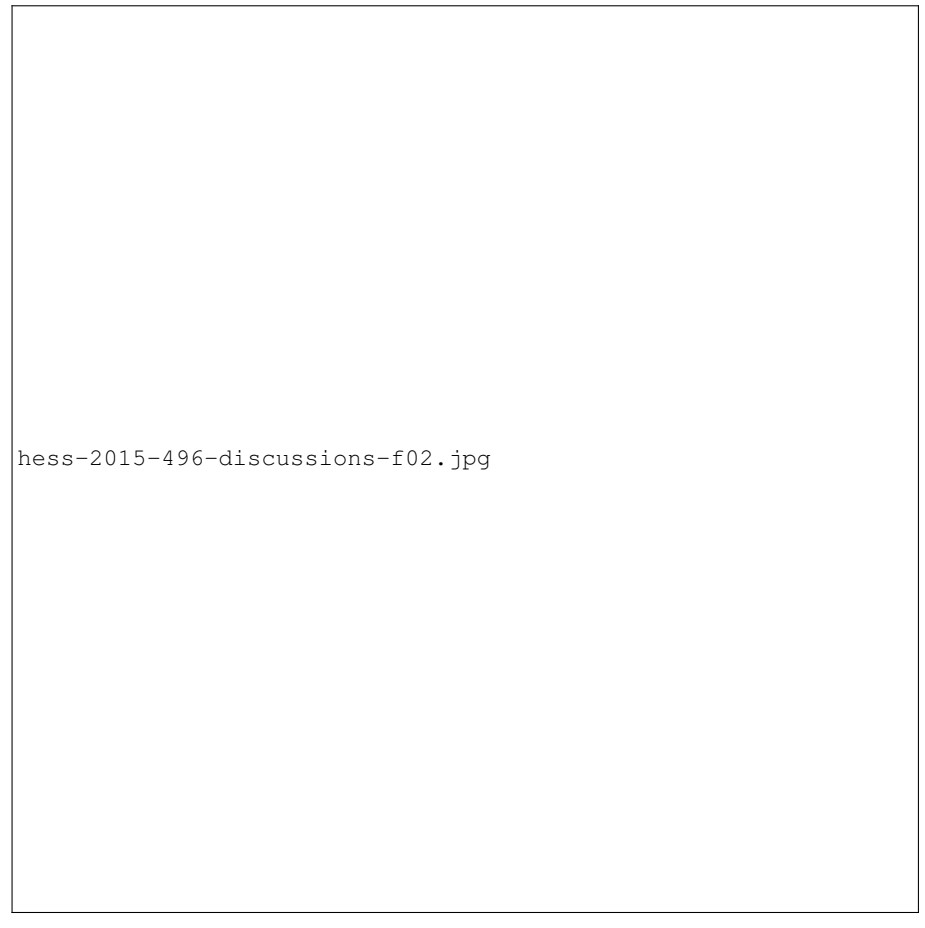

**Figure 2.** Designing and testing disdrometers (either with oil or flour) in a classroom with artificial drops of tinted water.

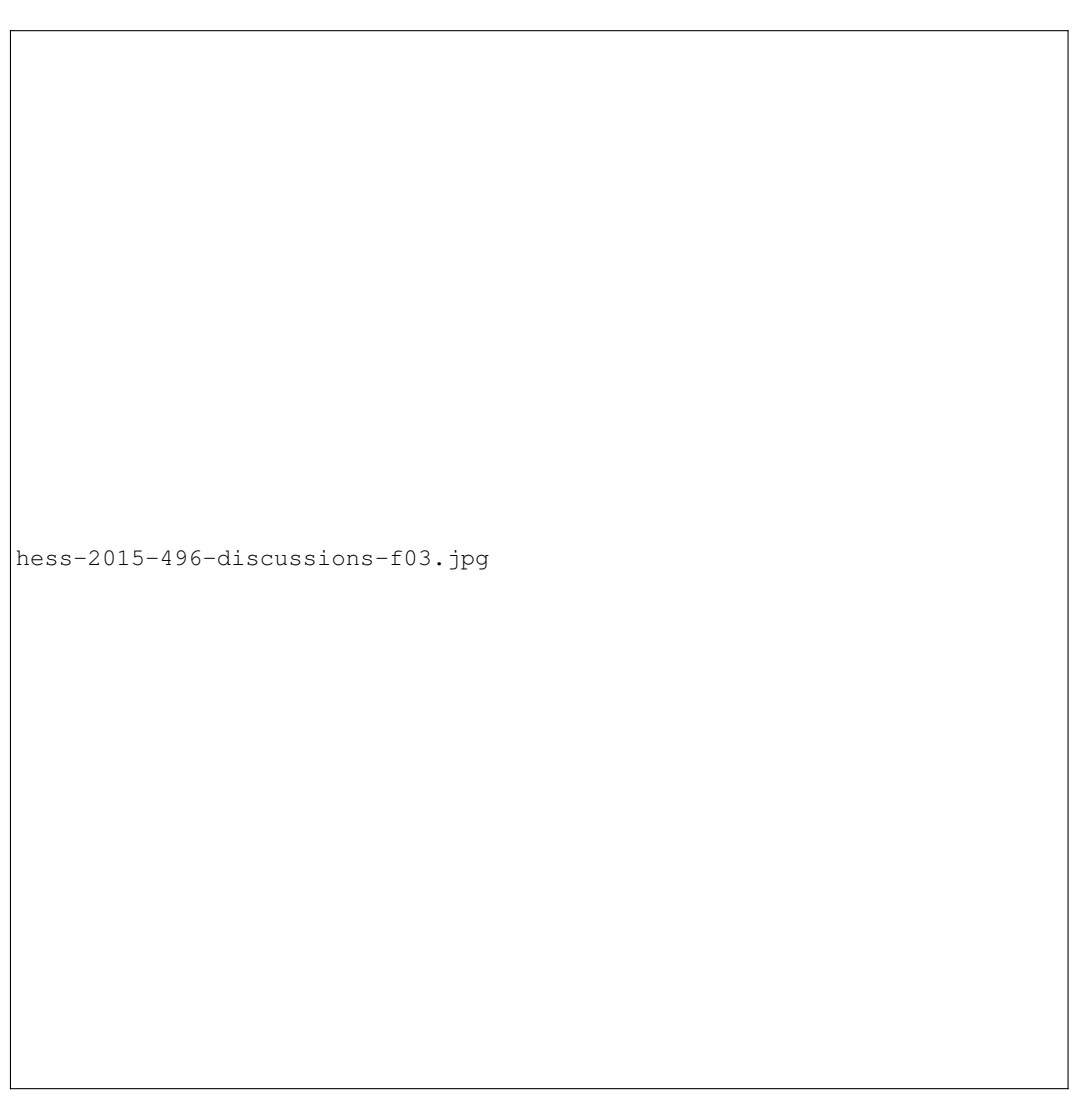

**Figure 3.** Use of the flour disdrometer in rainy conditions. **(a)** Experimenter bringing the device under rain and uncovering it few seconds. **(b)** Example of outcome where the various sizes of drops are visible. **(c)** Drawings by the children of their observations.

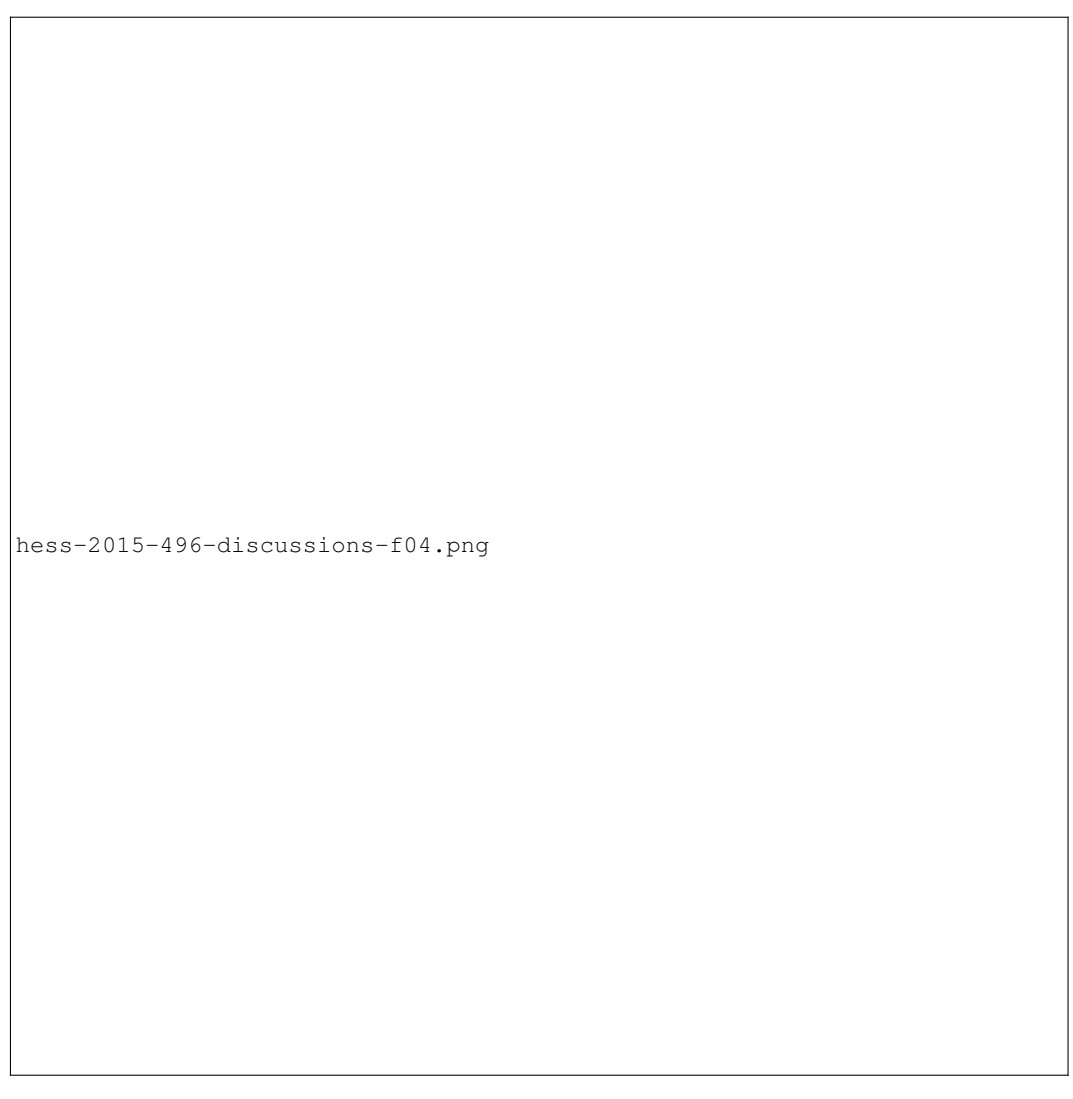

**Figure 4.** Representation of drop by drop data collected by a 2D-video-disdrometer during an event that occurred on 24 September 2012 in Ardèche (France). Each plot corresponds to 1 s and the corresponding time is indicated above it. The size of the sampling area is 11 cm × 11 cm. Raw data provided by Laboratoire de Télédétection en Environnement of Ecole Polytechnique Fédérale de Lausanne.

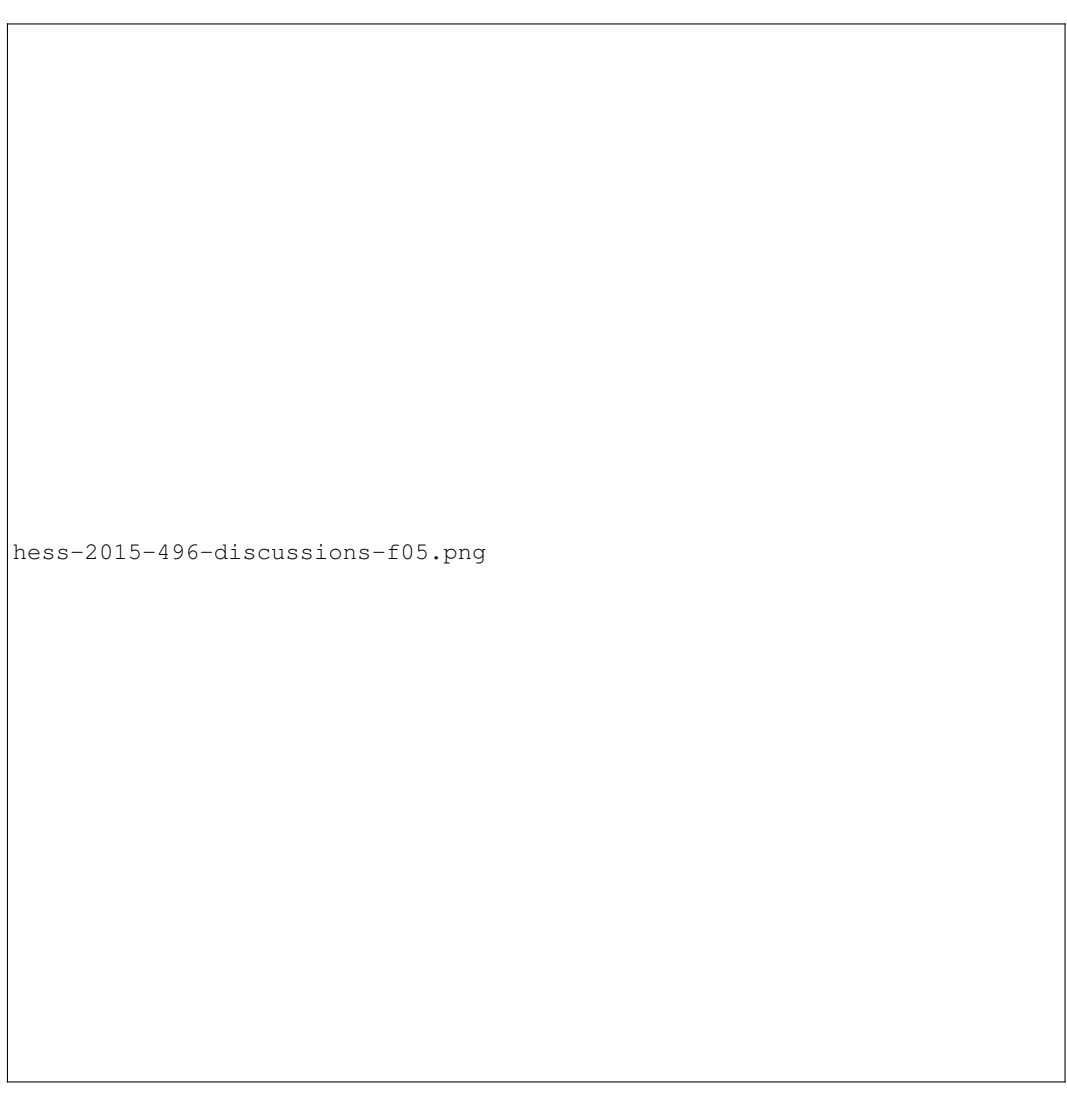

**Figure 5. (a)** Illustration of the pedagogical case of discrete multiplicative cascades. **(b)** Illustration of activity designed to mimic the specific case of the $\beta$-model cascade.

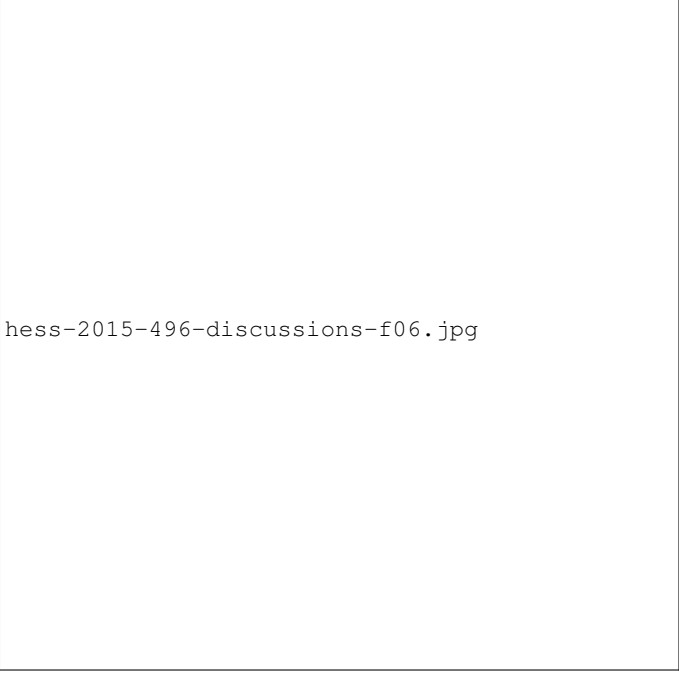

**Figure 6.** Daily time series of rainfall occurrence recorded by a class of 5–6 year old children in Sceaux (France) in October–November 2013.

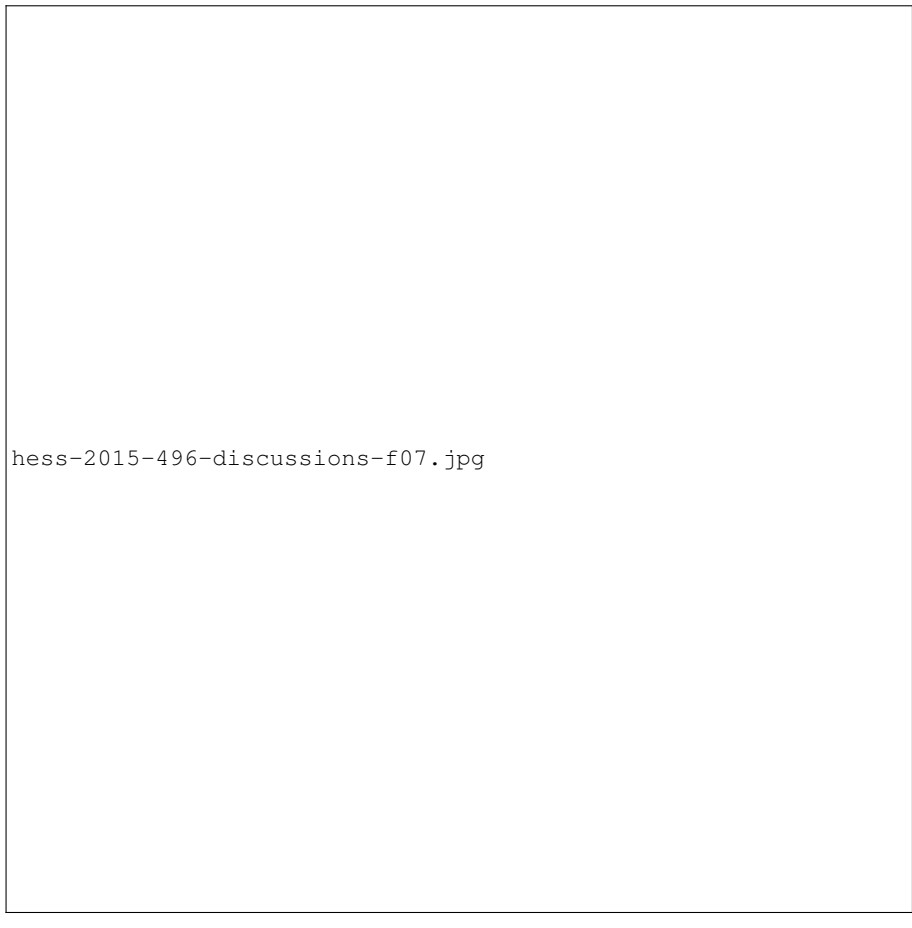

**Figure 7.** Implementation of the activity mimicking the $\beta$-model in a class of 5–6 year old children in Sceaux (January 2014).

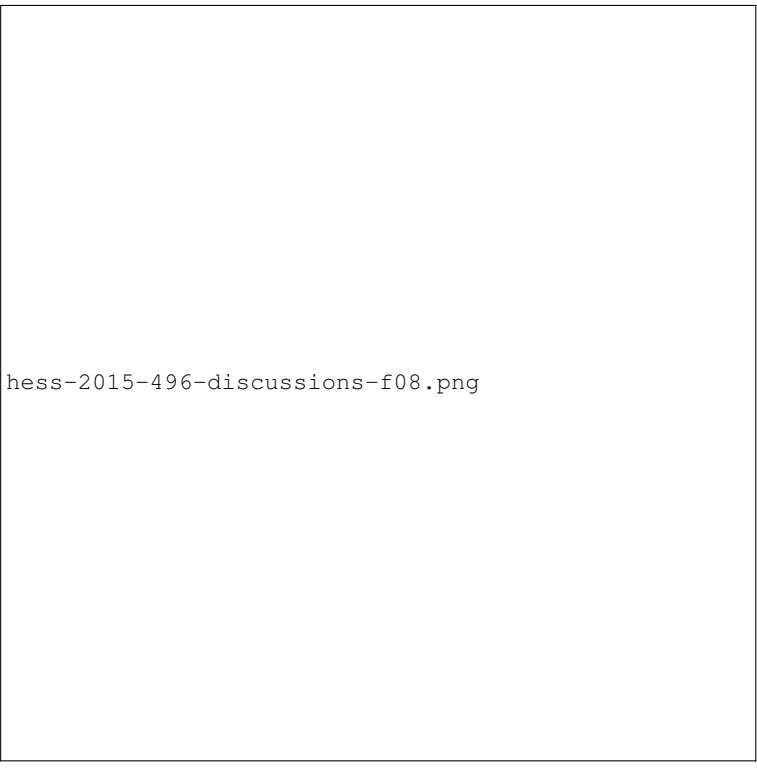

**Figure 8.** Examples of daily time series generated by the children with the $\beta$-model scheme.