# Peer review of "Making rainfall features fun: scientific activities for teaching children aged 5-12 years"

_Hydrology and Earth System Sciences, 2015_

## Referee Comment (RC1) · R. Hut (Referee) · 9 Feb 2016

The authors describe their effort to develop educational activities for children, both a "do an experiment" activity for kids aged 5-6 and a "write a book together with a scientist for kids aged 8-9. Whether the goals set out with activities are met is not evaluated, the paper merely describes the activities themselves. However, the design of the activities is based in a very extensive and commendable literature review. I would go as far as to say that this paper constitutes a very important review of the literature on effective science lessons in (primary) education, illustrated with the case study of developing a rainfall-education package. Given how many (geo)scientist develop, or consult in the development of, educational packaged at some point of their career, I judge this paper to be highly important for the (geo)scientific readership.
[Figure]

Having said that, I do have a few remarks:

- Can the authors explain why they did not include a quantitative evaluation in the first place? (page 17, line 10). Although, as explained above, the article stands on it's own as a literature review plus case study, but would have gained in value from a quantitative analyses that tests wether the goals set out in the design are indeed achieved. This is my main concern / comment on this paper. Further comments are minor.

- The first activity is done with a group of children aged 5-6 and the second with a group aged 8-9. In the introduction, the authors mention that the interest in school declines significantly at ages 11-14 (page 3, line 21). I'd like to ask the authors to elaborate how the choice of age groups that they made relates to this. Are the age groups chosen the most effective, if the goal is to interest more kids in (geo)science?

- I would strongly advice against (over)using Chinese proverbs (or other cultural "true-isms"). (page 4, line 1). My reading of the work cited at the top of page 4 is that offering different teaching modes is better for retention, not that any specific teaching mode ("involve me") is better than an other ("tell me"), merely that a mix of modes works best. Note that I do not advice to use "learning styles", but that what I take home from the articles cited is that offering a varied collection of experiences is best for retention.

- it would be helpful, for me, if the learning goals of the activities were mentioned in a central place, maybe in a table. Now they are scattered throughout the article (page 7 line 22 till page 8 line 4, page 7 line 15, page 9 line 1-5, etc.). I also believe that the authors mainly focus on knowledge transfer as a learning goal: the pupils should know about stuff at the end. However, I also believe that the actual skill involved in doing a measurement is worth mentioning as a
learning goal: the empowering notion that you can know something by measuring it yourself, in stead of trusting the knowledge passed on to you be others.

- The mathematical explanation of the cascade model is very detailed, for a paper that does not focus on the mathematics, but on education. Maybe the details of the model can be better mentioned in an appendix. Furthermore, I suggest to state that although the original model used "alive" and "dead" labels, in the case of this research, "wet" and "dry" will be used.

- in our review paper on "geoscience on tv", we included a paragraph on narrative structure. Maybe some of the references in that paragraph can be included on page 15. (The review paper is currently under review in HESSD: www.hydrol-earth-syst-sci-discuss.net/hess-2015-518/ )

- on page 16, lines 8-12, please indicate the qualitative nature of test to see if everything was understood.

- at some points, I noticed some mistakes in english, for example page 5, line 2 (the second "of" should go) and page 15, line 19 (the "the" should go). Since I am not a native English speaker, I may have missed additional mistakes in English and I advice to have the article proofread by a native speaker who wasn't involved in the article until now.

and some more personal notes:

- thanks for the reference to Maltese and Tai (2010), your lines 24-26 on page 3 helped me understand my own motivation to go into science. I also had one of these "specific memorable activities" in my primary education.

- the work of Som et al 2012, on 2.7 billion years old drop size distributions, is new to me and very cool!

Rolf Hut

disclaimer 1: the authors received funding in the RainGain project. In the group that I work in, we also received funding from RainGain. Personally I have never worked with (nor have I met, to my knowledge) any of the authors of this paper.

disclaimer 2: my own area of research is mainly sensor design in the hydrosphere. Although I have practical experience in science communication and teaching, I do not consider the "science of effective education" to be my specialty. I can therefore only gauge the hydrological part, and the general scientific soundness, of this paper as an expert. I will, for example, not be able to judge if the authors missed a key publication in the field of education that is essential to this paper.

---

## Referee Comment (RC2) · Anonymous Referee #2 · 19 Mar 2016

The authors present several different classroom-based activities around the core theme of rainfall. These activities include making flour and oil disdrometers, developing a rainfall times-series, and writing a children's book. These activities are part of the development of "a whole activity kit" on rainfall. Age groups engaged include 5-6 and 8-12 year olds. The activities described are interesting and appear to make complex topics accessible and fun for students as young as 5-6 years old.

The framework informing the design of the activities is strongly underpinned by the current literature and the authors use a mix of different methods to explore the topic of rainfall with students. The authors provide a review of relevant literature, which I feel is of great use to members of scientific community who engage in classroom-based outreach and education. Overall, this article provides a nice summary of detailed, scientist-led outreach and provides great examples for engaging students in the scientific process, through fun, real-life examples of the scope and rigor of research science.

Major Comments:

- One limitation of this work is the lack of qualitative or quantitative evaluation. This limits our understanding of the success and reach of these activities. Given the goals stated at the start of this manuscript, even qualitative evaluation would have helped to assess whether these goals were met. - Along the lines of evaluation, can the authors please provide in the text, (a) specific examples of how the teachers with whom they collaborated improved their activities (e.g. what language did they change for clarity etc.; lines 1-7, page 17) and; (b) examples of how and what feedback was collected from the students in the development of the book (line 8, page 16)

This information can guide readers in understanding how feedback was collected and the results of this work as well as demonstrate better one of the central arguments at the end of the study which states that, "The development of these activities highlighted the importance of a genuine collaboration between scientists and school teachers, which turns out to be necessary for a successful implementation" (Lines 1-3, page 17). If this is the case, please include a section about this collaboration (how connections were made and how did you build the dialogue, how was feedback solicited, examples of suggestions made by the educators) at the beginning of the article. It would also help to include specific information/examples about this feedback/collaboration for each of the activities presented in the manuscript.

Minor comments: - The article requires a thorough review of grammar and punctuation, including the use of the colon. - For clarity and ease of reading, the title can be simplified or clarified. I suggest removing (fractal!) from the first part of title as it only represents one small portion of the science of rainfall covered in the activities presented here and makes it hard to read. - I would specify an age range for your targeted activities in your abstract (page 2, line 3) and title (as above); young is vague and has different meanings in different countries - Page 2, line 6: I suggest changing fields to

Interactive
comment

science - First 10 lines in abstract and introduction are identical. . . - Page 2, line 9: I suggest removing the colon and specifically listing the activities that will be presented in the text e.g. (a) designing and using a disdrometer; (b) developing a time-series of local rainfall. . . and (c) collaboratively writing a children's book about rainfall. . . - Page 2 line 20: You mention parents here but this is not discussed in the text; remove or include more information in the text. - Page 2, line 26: what is Veolia? - Page 3, line 1: Please remove the colon and use sentences. - Page 3, line 9: use of 'young age' again; please define and be consistent. - Page 3, line 21: The ages mentioned in the text differs from that targeted with your activities. A majority of the activities presented here were targeted at 5-6 and 8-9 year olds but the ages cited in your literature review are much older (11-14 years old). Please elaborate on your choice of age range given that it differs from the literature, specifically if your stated goal was to interest more students in science. - Page 4, line 1: I would avoid using a Chinese proverb here. - Page 4, line 2: Not sure you need 19 references here. Also, please check the ordering of references required by the journal (e.g. chronological, alphabetical), and update this list accordingly. - Page 4, lines 11-12: Please provide references for this broad statement. If this is the assumption of the authors, please state that clearly. - Page 4, line 14: You start the sentence with "The Authors" but the cite numerous papers; please clarify the cited research from the opinion of the authors. - Page 5, line 9: Please state where geographically this, and all of the other activities, were tested. - Page 5, line 16: add a space between the 'R, the' - Page 5, line 17: reference? - Page 5, line 24: How does this activity differ from Mason and Viñas, 2013? Please describe. - Page 6, line 3: I really like this activity and how concepts were built for the students; I want to try this in the classroom! - Page 7, line 7: So it has to be raining to conduct this activity? Have you tested ways to 'make' rain (e.g. spray bottles)? - Pages 7-8, line 20: For the four points explored here, what materials are used to discuss or teach these elements? Up to this point, students are exploring the rainfall, not its formation. Can you describe the tools and resources used for this part of the activity? - Page 8, line 5: Do students measure the sizes? - Page 8, line 12: Remove () and add a comma - Page 8, line

13: Remove information about the plot and add it to the figure caption. - Pages 9-10, starting line 24: I suggest removing this 'footnote' reference - Page 10 (all)- 11 (lines 1-10): This information about the mathematics of the model is too detailed for this type of paper; I suggest removing it completely or moving it to an appendix. The information presented should be specific to the concepts explored with the students. Demonstrate the specifics when discussing the activity. - Page 11, line 11: I really like this activity, too. Fantastic! - Page 11, line 18: Change 'at school' to 'to school' - Page 11, line 23: holiday? - Page 12, lines 25-26: What is alive and dead in this context? First use, please explain. - Page 13, line 21: How do you know it "went well". What does that mean and how did you arrive at that conclusion? See major comments above. Here is a place where you can clearly state, what worked, what didn't and how you know. - Page 14, section 3.3: This seems out of context and is hard to understand. Some of this information is new and seems out of context. Did you explore these concepts in the classroom? If so, explain. If not, I suggest updating the text or section, as it does not demonstrate the students 'going further'. Is this a shortcoming of the activity? If so, that is really interesting and should be discussed in simple terms. Maybe each 'Going Further' section should be changed to focus on lessons learned or something similar? - Page 14, major comment: It seems that the fractal activity didn't work as anticipated; this is glossed over in the text. If this is the 'key' activity (as the title currently suggests), we need more information about what did and didn't work in this activity. The authors can go further here to describe what didn't work. This would be far more useful information than section 3.1. Given that the topic is complex, specific examples, in clear language, about how exactly this content was approached would be really useful (and good for the broader community). One key challenge is 'distilling' the science – what would the authors do differently? How did the expert educators help frame the content? - Page 14, line 20: Sceaux, France? - Page 14-15, starting line 26: Simplify text to limit punctuation. - Page 15: An interactive session implies two-way dialogue; please describe the design of the sessions- exactly how were they interactive. A bit more information would be useful if people wish to use a similar approach. - Page

15, line 16: Remove "This scientist writes the book". - Page 15, line 25: As above, I would list the questions so they are clear. e.g. The story was structured around three main questions, (a) xx; (b) xx ; (c) xx. . . . - Page 16, line 1: complements? - Page 16, point iii: What specific feedback did the children supply? What questions did you ask. Please describe. - Page 16: About the book development: who read the book, how was it distributed, what evaluation or metrics exist? What languages is it available in? - Page 16-17: Do you have any information about whether your goals were reached (e.g. student or teacher feedback?) - Page 17, line 13: You mention 'fruitful discussions' with the schools and teachers. Can you weave in specifics about these discussions into the text for each activity (as above with major comments)? - Pages 22-29: Check language and grammar in all figure captions. - Page 22: Keep caption formatting consistent; change (c) to read: (c) Student drawing of their observations. - Page 23: Change wording for clarity; e.g. testing disdrometers. . . - Page 24: Change to 'rainy conditions' and 'bringing the disdrometer outside to test it in the rain'; 'drawing by the children. . .'

––––––––––––––––––––––––––––––

---

## Author Comment (AC1) · 13 Apr 2016

First the authors would like to thank the reviewers for their suggestions that helped improve the manuscript. Hopefully the changes implemented will satisfy their requirements.

"Referee Comment 1

R. Hut (Referee) r.w.hut@tudelft.nl

The authors describe their effort to develop educational activities for children, both a "do an experiment" activity for kids aged 5-6 and a "write a book together with a scientist for kids aged 8-9. Whether the goals set out with activities are met is not evaluated, the paper merely describes the activities themselves. However, the design of the activities

is based in a very extensive and commendable literature review. I would go as far as to say that this paper constitutes a very important review of the literature on effective science lessons in (primary) education, illustrated with the case study of developing a rainfall-education package. Given how many (geo)scientist develop, or consult in the development of, educational packaged at some point of their career, I judge this paper to be highly important for the (geo)scientific readership.

Having said that, I do have a few remarks: • Can the authors explain why they did not include a quantitative evaluation in the first place? (page 17, line 10). Although, as explained above, the article stands on it's own as a literature review plus case study, but would have gained in value from a quantitative analyses that tests wether the goals set out in the design are indeed achieved. This is my main concern / comment on this paper. Further comments are minor."

The main reason, is that it was initially done as an activity in the class of the eldest son of the first author. It is now mentioned in the introduction. Future implementation will include a quantitative evaluation. Nevertheless some qualitative evaluation was included throughout the text, as also suggested by the other referee.

"• The first activity is done with a group of children aged 5-6 and the second with a group aged 8-9. In the introduction, the authors mention that the interest in school declines significantly at ages 11-14 (page 3, line 21). I'd like to ask the authors to elaborate how the choice of age groups that they made relates to this. Are the age groups chosen the most effective, if the goal is to interest more kids in (geo)science?"

This was also pointed out by the other referee. First the book is designed for children aged 8-12. It was done in a collaborative way with a class of children aged 8-9 so that it can be understandable for the whole age range targeted. With regards to the other activities, it is true that they were initially implemented with children aged 3-5 years. This was a practical choice driven by the fact that the son of one of the first author was in that class, which facilitated the first contact with the teacher ! This point was clarified

in the introduction. Since then the disdrometer experiment has been implemented in other place.

• I would strongly advice against (over)using Chinese proverbs (or other cultural "true-isms"). (page 4, line 1). My reading of the work cited at the top of page 4 is that offering different teaching modes is better for retention, not that any specific teaching mode ("involve me") is better than an other ("tell me"), merely that a mix of modes works best. Note that I do not advice to use "learning styles", but that what I take home from the articles cited is that offering a varied collection of experiences is best for retention.

The Chinese proverb was removed in the revised version of the paper. The referee is correct, and the paragraph was slightly changed to reflect more precisely this point.

"• it would be helpful, for me, if the learning goals of the activities were mentioned in a central place, maybe in a table. Now they are scattered throughout the article (page 7 line 22 till page 8 line 4, page 7 line 15, page 9 line 1-5, etc.). I also believe that the authors mainly focus on knowledge transfer as a learning goal: the pupils should know about stuff at the end. However, I also believe that the actual skill involved in doing a measurement is worth mentioning as a learning goal: the empowering notion that you can know something by measuring it yourself, in stead of trusting the knowledge passed on to you be others."

Authors have the feeling that the learning goals are rather visible at the beginning of each section. However if the referee wants them included in a table, it can easily be done. The "new" learning goal suggested by the referee is now mentioned in the discussion on the disdrometer activity.

"• The mathematical explanation of the cascade model is very detailed, for a paper that does not focus on the mathematics, but on education. Maybe the details of the model can be better mentioned in an appendix. Furthermore, I suggest to state that although the original model used "alive" and "dead" labels, in the case of this research, "wet" and "dry" will be used. "

Following the referee's suggestion, this portion was moved to an appendix for the interested reader. In the appendix the wording dead or alive was kept for historical reasons but your point mentioned.

"• in our review paper on "geoscience on tv", we included a paragraph on narrative structure. Maybe some of the references in that paragraph can be included on page 15. (The review paper is currently under review in HESSD: www.hydrol-earth-syst-sci-discuss.net/hess-2015-518/ )"

Thank you for this suggestion of interesting paper. We included a reference to it along with one to Dahlstrom (2014)

"• on page 16, lines 8-12, please indicate the qualitative nature of test to see if everything was understood."

A paragraph was added in section 3.2 to indicate precisely what work and what did not work.

"• at some points, I noticed some mistakes in english, for example page 5, line 2 (the second "of" should go) and page 15, line 19 (the "the" should go). Since I am not a native English speaker, I may have missed additional mistakes in English and I advice to have the article proofread by a native speaker who wasn't involved in the article until now."

The article was proofread again.

"and some more personal notes: • thanks for the reference to Maltese and Tai (2010), your lines 24-26 on page 3 helped me understand my own motivation to go into science. I also had one of these "specific memorable activities" in my primary education. • the work of Som et al 2012, on 2.7 billion years old drop size distributions, is new to me and very cool!"

Thanks !

---

## Author Comment (AC2) · 13 Apr 2016

First the authors would like to thank the reviewers for their suggestions that helped improve the manuscript. Hopefully the changes implemented will satisfy their requirements.

"Referee Comment 2

The authors present several different classroom-based activities around the core theme of rainfall. These activities include making flour and oil disdrometers, developing a rainfall times-series, and writing a children's book. These activities are part of the development of "a whole activity kit" on rainfall. Age groups engaged include 5-6 and 8-12 year olds. The activities described are interesting and appear to make complex topics accessible and fun for students as young as 5-6 years old. The framework

informing the design of the activities is strongly underpinned by the current literature and the authors use a mix of different methods to explore the topic of rainfall with students. The authors provide a review of relevant literature, which I feel is of great use to members of scientific community who engage in classroom-based outreach and education. Overall, this article provides a nice summary of detailed, scientist-led outreach and provides great examples for engaging students in the scientific process, through fun, real-life examples of the scope and rigor of research science.

Major Comments: - One limitation of this work is the lack of qualitative or quantitative evaluation. This limits our understanding of the success and reach of these activities. Given the goals stated at the start of this manuscript, even qualitative evaluation would have helped to assess whether these goals were met. - Along the lines of evaluation, can the authors please provide in the text, (a) specific examples of how the teachers with whom they collaborated improved their activities (e.g. what language did they change for clarity etc.; lines 1-7, page 17) and; (b) examples of how and what feedback was collected from the students in the development of the book (line 8, page 16)"

This information can guide readers in understanding how feedback was collected and the results of this work as well as demonstrate better one of the central arguments at the end of the study which states that, "The development of these activities highlighted the importance of a genuine collaboration between scientists and school teachers, which turns out to be necessary for a successful implementation" (Lines 1-3, page 17). If this is the case, please include a section about this collaboration (how connections were made and how did you build the dialogue, how was feedback solicited, examples of suggestions made by the educators) at the beginning of the article. It would also help to include specific information/examples about this feedback/collaboration for each of the activities presented in the manuscript.

As suggested by the referee, qualitative evaluation was included throughout the discussion in the text and also elements on the joint work with teachers

"Minor comments: - The article requires a thorough review of grammar and punctuation, including the use of the colon. -

This was done.

"- For clarity and ease of reading, the title can be simplified or clarified. I suggest removing (fractal!) from the first part of title as it only represents one small portion of the science of rainfall covered in the activities presented here and makes it hard to read."

As suggested by the referee, fractal was removed from the title in the revised version of the manuscript

- I would specify an age range for your targeted activities in your abstract (page 2, line 3) and title (as above); young is vague and has different meanings in different countries

This was done.

- Page 2, line 6: I suggest changing fields to science

The sentence was changed according to your suggestion.

- First 10 lines in abstract and introduction are identical. . . - Page 2, line 9: I suggest removing the colon and specifically listing the activities that will be presented in the text e.g. (a) designing and using a disdrometer; (b) developing a time-series of local rainfall. . . and (c) collaboratively writing a children's book about rainfall. . .

This was changed.

- Page 2 line 20: You mention parents here but this is not discussed in the text; remove or include more information in the text.

It was mentioned because the activity was actually implemented in the class of a son of the first author and reported on the class blog. This point has been added in the conclusion:

"Finally it should be mentioned that the activities done with children aged 5-6 years were reported on the blog's classroom handled by the teacher. It is a non-public blog accessible only to the parents because it contains pictures of the kids while they are at school. Since they were implemented in the class of the first author son, the publication triggered some unusual and pleasant discussions on drop size distribution at the end of school. Indeed parents were not aware of the variety of drop size as well and wanted to learn more. Some of them actually tested the experiment at home with their child. The first authors used these short discussions to make parents aware of the extreme variability of rainfall as well as the difficulty to properly measure it. This reflects nicely how an activity designed for young children actually ends up with parents learning new thing as well."

- Page 2, line 26: what is Veolia?

It is a water company. A link to their website was added.

- Page 3, line 1: Please remove the colon and use sentences.

For clarity the three activities are now presented as a list.

- Page 3, line 9: use of 'young age' again; please define and be consistent.

The paper by Sackes in targeted to children aged 3-5 years. This was added in the manuscript.

"- Page 3, line 21: The ages mentioned in the text differs from that targeted with your activities. A majority of the activities presented here were targeted at 5-6 and 8-9 year olds but the ages cited in your literature review are much older (11-14 years old). Please elaborate on your choice of age range given that it differs from the literature, specifically if your stated goal was to interest more students in science."

First the book is designed for children aged 8-12. It was done a collaborative way with a class of children aged 8-9 so that it can be understandable for the whole age range targeted. With regards to the other activities, it is true that they were initially imple-

mented with children aged 3-5 years. This was a practical choice driven by the fact that the son of one of the first author was in that class, which facilitated the first contact with the teacher ! This point was clarified in the introduction. Since then the disdrometer experiment has been implemented in other place and this is now discussed in section 2.

- Page 4, line 1: I would avoid using a Chinese proverb here.

It was removed in the revised version of the paper.

- Page 4, line 2: Not sure you need 19 references here. Also, please check the ordering of references required by the journal (e.g. chronological, alphabetical), and update this list accordingly. - Page 4, lines 11-12: Please provide references for this broad statement. If this is the assumption of the authors, please state that clearly. - Page 4, line 14: You start the sentence with "The Authors" but the cite numerous papers; please clarify the cited research from the opinion of the authors. - Page 5, line 9: Please state where geographically this, and all of the other activities, were tested.

This was done.

- Page 5, line 16: add a space between the 'R, the'

This was done, thank you for your careful reading.

- Page 5, line 17: reference?

This is actually in the same paper Marshall and Palmer (1948). This was clarified in the manuscript.

- Page 5, line 24: How does this activity differ from Mason and Viñas, 2013? Please describe.

The experiment is similar, except that Mason and Viñas, 2013 implemented it for high-school students. Hence they developed it on a larger area and also introduced the measurement of drop size and not just visual inspection. They also only tested the

"flour" disdrometer and did not tried the "oil" one. The reference was actually moved to the following section.

- Page 6, line 3: I really like this activity and how concepts were built for the students; I want to try this in the classroom!

Thanks !

- Page 7, line 7: So it has to be raining to conduct this activity?

Yes ! The following sentence was added in the corresponding section : "We were lucky that it was actually raining the day of the experiment. In case it had not been, some pictures were ready to continue the discussion and the teacher would have done the actual test once some rain appeared."

- Have you tested ways to 'make' rain (e.g. spray bottles)?

No, that could be something to test for future improvements.

- Pages 7-8, line 20: For the four points explored here, what materials are used to discuss or teach these elements? Up to this point, students are exploring the rainfall, not its formation. Can you describe the tools and resources used for this part of the activity?

The following paragraph was added : "For this part of the activity, no dedicated tools were used and it was only based on a discussion. To illustrate the first point, the standard example of a the condensation around a bottle taken out of a refrigerator was used and some kids recognize this effect. Some specific activities should be developed to address this issue in future works."

- Page 8, line 5: Do students measure the sizes?

No they were too young for this. The following sentence was added for clarification : "The experiment is similar, except that Mason and Viñas, 2013 implemented it for high-school students. Hence they developed it on a larger area and also introduced

the measurement of drop size and not just visual inspection. They also only tested the "flour" disdrometer and did not tried the "oil" one. The reference was actually moved to the following section."

- Page 8, line 12: Remove () and add a comma

This was done.

- Page 8, line 13: Remove information about the plot and add it to the figure caption.

This was done.

- Pages 9-10, starting line 24: I suggest removing this 'footnote' reference

Authors would rather keep it since we believe that it illustrates in a rather pedagogical way how cascade models were introduced to model turbulence.

- Page 10 (all)- 11 (lines 1-10): This information about the mathematics of the model is too detailed for this type of paper; I suggest removing it completely or moving it to an appendix. The information presented should be specific to the concepts explored with the students. Demonstrate the specifics when discussing the activity.

Following the referee's suggestion, this portion was moved to an appendix for the interested reader.

- Page 11, line 11: I really like this activity, too. Fantastic!

Thank you, if you are interested in potential collaboration, feel free to contact us privately !

- Page 11, line 18: Change 'at school' to 'to school'

This was done.

- Page 11, line 23: holiday?

Yes, the spelling was not correct, it was changed.

- Page 12, lines 25-26: What is alive and dead in this context? First use, please explain.

In this context, it means "dry" and "rainy". The initial words were kept for historical reasons but your point is now mentioned.

- Page 13, line 21: How do you know it "went well". What does that mean and how did you arrive at that conclusion? See major comments above. Here is a place where you can clearly state, what worked, what didn't and how you know.

Following the referee's comment, the end of the section was re-written and completed to explain more precisely what worked and what did not work.

- Page 14, section 3.3: This seems out of context and is hard to understand. Some of this information is new and seems out of context. Did you explore these concepts in the classroom? If so, explain. If not, I suggest updating the text or section, as it does not demonstrate the students 'going further'. Is this a shortcoming of the activity? If so, that is really interesting and should be discussed in simple terms. Maybe each 'Going Further' section should be changed to focus on lessons learned or something similar?

The presentation of each activity, is preceded by an historical and scientific background section and followed by a 'going further' section. The purpose of these sections is to provide teachers and educators with sufficient material so that they feel comfortable when implementing the activity. According to the age of the children they will use or not this additional information. The activities were initially implemented in a classroom with children. This was clarified in the introduction.

So the section was kept (with slight modifications). Nevertheless, following the referee's comment we added a paragraph to suggest ideas on how to introduce these concept with older kids (5-6 years is likely to be too young to start).

- Page 14, major comment: It seems that the fractal activity didn't work as anticipated; this is glossed over in the text. If this is the 'key' activity (as the title currently suggests), we need more information about what did and didn't work in this activity. The

authors can go further here to describe what didn't work. This would be far more useful information than section 3.1. Given that the topic is complex, specific examples, in clear language, about how exactly this content was approached would be really useful (and good for the broader community). One key challenge is 'distilling' the science – what would the authors do differently? How did the expert educators help frame the content?

The "fractal activity" is not the especially the central one, and as you suggested the title was changed. A precise discussion on what worked and what did not was added in section 3.2 (see answer above)

- Page 14, line 20: Sceaux, France?

Yes, it was clarified.

- Page 14-15, starting line 26: Simplify text to limit punctuation.

This was done, shorter sentences are now used.

- Page 15: An interactive session implies two-way dialogue; please describe the design of the sessions- exactly how were they interactive. A bit more information would be useful if people wish to use a similar approach.

The section was largely expended to provide the reader with more details.

- Page 15, line 16: Remove "This scientist writes the book".

The sentence was changed to : "The book is then written by the scientist, with the aim of answering (at least partially) all the questions raised by children."

- Page 15, line 25: As above, I would list the questions so they are clear. e.g. The story was structured around three main questions, (a) xx; (b) xx ; (c) xx. . ..

This was done.

- Page 16, line 1: complements? - Page 16, point iii: What specific feedback did the

children supply? What questions did you ask. Please describe.

Following the referee's comment, the section was completed : "The main point was to ask them if they had understood everything was, and whether they had some suggestions regarding to the characters. They were happy with the characters and had only minor suggestions for the content. For example the explanation of a rainbow effect was re-written. Actually they had more questions on the process of a book creation (how many people worked on it ? How long does it take ? How is it printed ? ...), which the editor answered. After this session, the scientist made some minor adjustments to improve the book."

- Page 16: About the book development: who read the book, how was it distributed, what evaluation or metrics exist? What languages is it available in? - Page 16-17: Do you have any information about whether your goals were reached (e.g. student or teacher feedback?)

A short paragraph was added at the end of the section: "The book is then made available to the public (bookshop, internet...). Typically 2000 to 3000 are sold in this collection over the life of the book. It has not yet been translated to other languages. Authors did not get a very precise feedback from the teacher apart from the fact that they were satisfied with the experience. The two interactive session were dynamic which shows the interest of the children in the activity."

- Page 17, line 13: You mention 'fruitful discussions' with the schools and teachers. Can you weave in specifics about these discussions into the text for each activity (as above with major comments)?

Some elements were included throughout the manuscript to make this point clearer.

"- Pages 22-29: Check language and grammar in all figure captions. - Page 22: Keep caption formatting consistent; change (c) to read: (c) Student drawing of their observations. - Page 23: Change wording for clarity; e.g. testing disdrometers. . . - Page 24:

Change to 'rainy conditions' and 'bringing the disdrometer outside to test it in the rain'; 'drawing by the children. . ."

This was done.